# Pyrolysis Kinetic Study of Polylactic Acid

**DOI:** 10.3390/polym15010012

**Published:** 2022-12-20

**Authors:** Zaid Alhulaybi, Ibrahim Dubdub, Mohammed Al-Yaari, Abdulrahman Almithn, Abdullah F. Al-Naim, Haidar Aljanubi

**Affiliations:** 1Chemical Engineering Department, King Faisal University, P.O. Box 380, Al-Ahsa 31982, Saudi Arabia; 2Physics Department, King Faisal University, P.O. Box 400, Al-Ahsa 31982, Saudi Arabia; 3Electrical Engineering Department, King Faisal University, P.O. Box 380, Al-Ahsa 31982, Saudi Arabia

**Keywords:** pyrolysis, PLA, activation energy, thermogravimetric analyzer (TGA), kinetics, biodegradable polymer, recycling

## Abstract

Polylactic acid (PLA) is a biodegradable polymer and is mainly used in the textile and food packaging fields. The aim of this work is to build knowledge on the kinetics of the pyrolysis of PLA with the help of thermogravimetric analysis (TGA) using four model-free methods, namely Friedman, Flynn–Wall–Qzawa (FWO), Kissinger–Akahira–Sunose (KAS), and Starink. Additionally, two model-fitting methods (the Coats–Redfern and Criado methods) were applied. TGA data at 5, 10, 20, and 30 K/min heating rates were collected. The obtained activation energies of the pyrolysis of PLA at different conversions by the model-free models were in good agreement and the average values were 97, 109, 104, and 104 kJ/mol for Friedman, FWO, KAS, and Starink, respectively. The Criado model was used together with the Coats–Redfern model to identify the most appropriate reaction mechanism. As per this work, the best controlling reaction mechanism of the PLA pyrolysis can be expressed by the geometrical contraction model (R2).

## 1. Introduction

The rapidly growing human population is faced with the problem of non-biodegradable polymers. Non-biodegradable materials are commonly incinerated or buried under the soil, but both methods pose environmental problems because they release toxic gases and contaminate the air, water and soil. By replacing non-biodegradable polymers with biodegradable polymers and composites, the use of harmful non-biodegradable polymers can be reduced (Halász and Csóka (2013) [1]).

There are several types of biodegradable polymers including polylactic acids (PLA), chitosan, polycaprolactone acid (PGL), polyglycolic acid (PGA), polyvinyl chloride (PVA), and polyhydroxybutyrate (Halász and Csóka (2013) [1], Yuniarto et al. (2016) [2]). 

Polymers, such as PLA, are an admirable alternative to non-biodegradable polymers. PLA, as an example, is a thermoplastic aliphatic polyester produced via fermentation processes from renewable resources such as corn and starch. Incineration of PLA biopolymer does not produce any harmful gases. It has high stiffness, good strength, and is easy to process mold, and shape. When PLA degrades, it releases H_2_O and CO_2_. A PLA product can be extruded, blown molded, or poured into a solvent. In comparison with other polymers such as polyethylene glycol (PEG), polycaprolactone, and polyhydroxybuterate, PLA has higher thermal stability. Due to its thermal properties, PLA finds applications in the textile and food packaging industries (Dhar et al. (2015) [3]).

As well as being biodegradable, PLA has respectable mechanical properties when compared to other biodegradable polymers. Many applications have been developed for PLA including but not limited to, drug delivery systems, textiles, films/membranes, tissue engineering scaffolds, biological scaffolds, and others. There are several limitations associated with PLA, including low impact resistance at room temperature, poor oxygen and water barrier properties, as well as high rigidity and brittleness. The poor barrier properties of PLA films make them unsuitable for food packaging. These problems must be overcome for PLA to have a wider range of commercial applications. To a certain extent, PLA limitations can be overcome by combining them with plasticizers, reinforcement fillers, and other polymers. Mortezaeikia et al. (2021) [4] reviewed and presented comprehensively the kinetic pyrolysis models and methods including model-free methods and model-fitting methods of TGA data for all polymers. Saad et al. (2021) [5] studied mainly the effect pyrolysis atmosphere (nitrogen or carbon dioxide) on the pyrolytic cracking of six different polymers: high- and low-density polyethylene (HDPE and LDPE), polypropylene (PP), polystyrene (PS), and polyethylene terephthalate (PET). They showed only one peak reaction for all polymers with shifting to higher temperature in a carbon dioxide atmosphere. They found the activation energy values decreasing in the following order: LDPE > HDPE > PS > PP > PET in the nitrogen atmosphere, and: HDPE > LDPE > PET > PP> PS in the carbon dioxide atmosphere. Pan et al. (2021) [6] investigated the kinetic pyrolysis of the waste polyethylene (WPE) and PE using genetic algorithm (GA) and isoconversional methods with thermogravimetric analysis (TGA). They found that the reaction-order model was the most convenient model describing the pyrolysis among the extended Prout–Tompkins and Sestak–Berggren models.

Zhang et al. (2022) [7] studied the co-pyrolysis of lychee and plastic wastes (LPW) using TGA data at four different heating rates (10, 20, 30, and 40 K min^−1^). They found that the activation energy values ranged from 64 to 71 kJ mol^−1^, using the FWO, KAS, and Starink methods.

Soufizadeh et al. (2022) [8] used the kinetic approaches and the quality control techniques as new ways of estimating and optimizing the effect of plastic waste pyrolysis processes such as the conversion rate, heating rate, and pyrolysis temperature on the values of the activation energy. They concluded that the activation energy of the pyrolysis process would be minimized with the increase in the conversion rate, heating rate and with a reduction in temperature.

Generally, to examine the thermal degradation kinetics of polymeric blends, kinetic modeling using data gained from some thermal analysis techniques such as differential scanning calorimetry (DSC) and thermogravimetric analysis (TGA) can be extremely helpful. It is possible to estimate with precision the degradation reaction rate, as well as the mechanisms involved, by using the kinetic parameters obtained. Muravyev and Vyazovkin (2022) [9] reviewed the most proper options that should be chosen and as advised by the International Confederation for Thermal Analysis and Calorimetry (ICTAC). ICTAC mainly recommended using the model-free methods, where the data should be collected at more than three different heating rates. ICTAC emphasized that running TGA experiments should be according to its instructions in order to collect the proper data for kinetic parameters calculations. It recommended that the ratio between the highest and the lowest heating should be more than 10 (Osman et al. (2022) [10]). Since the TG curve shows only one peak reaction, single-step thermal decomposition can be considered to derive the proper kinetic parameters (Koga et al. (2023) [11]).

Using TGA analysis to obtain the pyrolysis kinetic parameters has grown in popularity over the years, and it has proven convincingly useful. Table 1 summarizes some of the research work studying the pyrolysis of PLA and its derived matrixes and different polymers.

This work aims to build knowledge on the kinetics of the pyrolysis of PLA non-porous films made through compression molding. TGA analysis at four different heating rates (5, 10, 20, and 30 K/min) was conducted. The kinetic triplets of the PLA pyrolysis were obtained using four isoconversional models (Friedman, FWO, KAS, and Starink) and two non-isoconversional models (Coats–Redfern, and Criado). This research project aimed to be extended further to study the effect of the incorporation of some biomass materials (agriculture wastes) in order to produce environmentally sustainable material(s) for the next generation. 

## 2. Materials and Methods

### 2.1. Material

PLA2003D semi-crystalline biopolymer (M_W_ ≈ 200 KDa) was supplied by Ingeo™ Biopolymer, Minnetonka, USA. Some of its physical properties are presented in Table 2. 

### 2.2. Proximate Analysis

Proximate analysis was made to determine the moisture, volatile matter, fixed carbon, and ash contents using the Simultaneous Q50 Thermal Analyzer manufactured by TA Instruments, USA. Results are presented in Table 3 and details of this analysis are fully described elsewhere (Dubdub and Al-Yaari (2020) [16]).

PLA pellets were melted at 190 °C for 15 min using a conventional oven. Subsequently, molten PLA cooled down at room temperature for 10 min. Synthesized PLA sheets were used for the TGA analysis.

### 2.3. Thermogravimetry 

PLA non-porous matrixes were carried out directly to the thermogravimetric analyzer. Forty mg of PLA samples were used throughout the study. TGA experiments were performed in an inert atmosphere of pure N_2_ (flowrate 60 mL/min) at four different heating rates (5, 10, 20, and 30 K/min).

### 2.4. Determination of the Kinetic Triplets 

Generally, the reaction rate (r) depends on conversion (α), temperature (*T*), and pressure (*P*). However, for the solid-state reactions, the pressure effect is negligible. Therefore, using the Arrhenius relationship, we found that the reaction rate equation of the PLA pyrolysis can be expressed as (Aboulkas et al. (2010) [17]): (1)$$-r=\frac{d\alpha }{dt}=A\exp\left(-\frac{E}{RT}\;\right)f\left(\alpha \right)\;$$
where: 

*t* is time,

A is the frequency factor,

E is the activation energy,

R is the universal gas constant, and

f(α) is the conversion-dependent term which depends on the reaction mechanism as presented in Appendix A. From the TGA experimental data, the reaction conversion can be obtained as a fraction of the PLA weight loss.

For the non-isothermal pyrolysis of PLA, the heating rate (β=dT/dt) can be included in the reaction rate equation to become (Khodaparasti et al. (2022) [18]): (2)$$\beta \frac{d\alpha }{dT}=A\exp\left(-\frac{E}{RT}\;\right)f\left(\alpha \right)\;$$
and thus:(3)$$g\left(\alpha \right)=\frac{A}{\beta }\int_{T_{o}}^{T}\exp\left(-\frac{E}{RT}\;\right)\;dT\;$$
where:$$g\left(\alpha \right)=\int_{0}^{\alpha }\frac{d\alpha }{f\left(\alpha \right)},\;\mathrm{and}$$

*T_o_* is the initial PLA pyrolysis temperature.

The temperature integral $\left[\int_{T_{o}}^{T}\exp\left(-\frac{E}{RT}\;\right)\;dT\right]$ does not have an analytical solution, but can be approximated as (Khodaparasti et al. (2022) [18]): (4)$$\int_{T_{o}}^{T}\exp\left(-\frac{E}{RT}\;\right)\;dT≅\frac{E}{\;R}\;P\left(-\frac{E}{RT}\;\right)\;$$

Researchers utilized different numerical methods and series expansions to approximate the polynomial term $\left[P\left(-\frac{E}{RT}\;\right)\right]$ and thus different models were proposed. Those models are used to obtain the kinetic triplets (A, E, and mechanism) (Aboulkas et al. (2010) [17], Duque et al. (2020) [19], Mumbach et al. (2019) [20], Al-Yaari and Dubdub (2020) [21]). Table 4 and Table 5 list four of the most used isoconversional models and two of the non-isoconversional models, respectively.

In this work, the isoconversional models (Friedman, FWO, KAS, and Starink) were used to obtain initially the activation energy of the PLA pyrolysis. Subsequently, the non-isoconversional models (Coats–Redfern and Criado) were used to determine the most appropriate reaction mechanism. Finally, the pre-exponential factor (*A*) was obtained by the isoconversional models.

## 3. Results and Discussion

### 3.1. TG-DTG Analysis 

TG and DTG at four different heating rates (5, 10, 20, and 30 K/min) of pyrolytic cracking of PLA are shown in Figure 1. Thermograms were similar for all different heating rates. As in the pyrolysis of other polymers [21,22,23], the pyrolytic temperature characteristics (T_onset_, T_peak_, and T_endset_) were shifted to higher temperatures as the heating rate raised. Additionally, it has been shown that PLA polymers degrade by only one reaction. Table 6 shows the onset, peak, and final temperatures of the pyrolytic degradation of PLA at different heating rates. Table 7 shows the reported peak temperatures for PLA pyrolysis at a 10 K/min heating rate [10,11,12,13]. In this work, the peak temperature was observed to be 647 K, which lies between the lowest (622 K) and the highest (657 K) reported temperatures. DTG curves show that the peak value decreased from 2.75 1/min at 5 K/min to 2 1/min at 30 K/min. 

### 3.2. Isoconversional Kinetics Models

Equation (1) is considered the startup equation from which all the models’ equations can be derived. Four isoconversional models have been used to calculate the main kinetic parameter (activation energy in kJ/mol) at conversions ranging from 0.1 to 0.9. Figure 2 shows the regression lines of the experimental data of the PLA pyrolysis by Friedman, FWO, KAS, and Starink models. Muravyev et al. (2019) [24] pointed to the importance of considering the recommendation of the kinetic committee of ICTAC. They mentioned that the activation energy values calculated by any of the isoconversional methods for a single-step reaction (integral method: FWO, or differential method: Friedman) should be ideally constant with the range values of the conversion. However, this is not the case in the real results, and this can be attributed to the changes in the reaction condition, reaction geometry, and rate limiting as the reaction moves from low to high conversion. Good linear relationships, with high values of *R^2^*, were obtained and the *E* values obtained by the isoconversional models are presented in Figure 3 and Table 8. The highest and lowest values of *E* were obtained by FWO, and Friedman, respectively. However, KAS and Starink gave the same value of *E*. The obtained values for all four isoconversional models are consistent with the reported values as presented in Table 9. There is somehow a trend in decreasing the *E* values by increasing the conversion from 0.1 to 0.9 for FOW, KAS, and Starink. However, for the Friedman method, the obtained values of *E* started high (115 kJ/mol at α = 0.1), then decreased as the conversion increased (89 kJ/mol at α = 0.5), and after that, they increased steadily (91 kJ/mol at α = 0.9). Mróz et al. (2013) [12] reported that the *E* values dependency on the conversion (between 0.1 and 0.9) has two regions, and thus the reaction mechanism changes as well. In addition, Aoyagi et al. (2002) [25] confirmed that the reaction of PLA pyrolysis occurred with more than a single mechanism. In addition, this finding is in respectable agreement with the results reported by Li et al. (2009) [26].

### 3.3. Model-Fitting Kinetics Methods

The first non-isoconversional method “Coats–Redfern” was used by plotting $\ln\left[\frac{g\left(\alpha \right)}{T^{2}}\right]$ versus 1/T for different reaction mechanisms (See Appendix A) and straight-line correlations were obtained. The most appropriate reaction mechanism can be determined by comparing the obtained *E* value (with high *R^2^*) by the Coats–Redfern method with those obtained by the isoconversional models. Table 10 presents the values of the activation energy attained by the Coats–Redfern models for 15 different reaction mechanisms and different values (with different values of *R^2^*). As per the results reported in Table 11, the obtained *E* value for the second Avrami–Erofeev (A2) and the geometrical contraction model (R2) were 97 kJ/mol (with a regression coefficient of 0.9991), and 119 kJ/mol (with a regression coefficient of 0.9999), respectively, which are the closest values to the ones obtained by the isoconversional models. Thus, the A2 and R2 reactions can be the most suitable mechanisms for PLA pyrolysis.

Furthermore, reaction mechanisms can also be evaluated by the Criado model as shown in Figure 4. Graphs of D4, R1, P2, P3, and P4 were excluded from Figure 4 because their curves are far from the experimental curves in all tests. As shown in Figure 4, as per the Criado model, although the A2 model did not fit the experimental data, the three-dimensional diffusion (D3), and the geometrical contraction cylinder (R2) models have the closest curves to the experimental ones. However, since the *E* value obtained by the Coats–Redfern for the D3 models (354 kJ/mol) is far from the obtained value by the isoconversional model, the D3 model is not suitable to represent the PLA pyrolysis. Table 11 shows the values of *E*, *ln(A)*, and *R*^2^ for each heating rate for the most suitable reaction mechanism (R2). Bhiogade et al. (2020) [13] used the Coats–Redfern model together with the Mampel (first-order reaction) model to determine the kinetic parameters. An *E* value of 79.21 kJ/mol was reported.

## 4. Conclusions

In this work, the PLA pyrolysis was investigated and non-isothermal TGA data were analyzed. Kinetic studies at 5, 10, 20, and 30 K/min using both isoconversional (Friedman, FWO, KAS, and Starink) and non-isoconversional (Coats–Redfern and Criado) models were performed. The obtained values of activation energy by both methods were comparable. The average *E* values found by the four isoconversional models at different conversions were in good agreement (Friedman: 97 kJ/mol, FWO: 109 kJ/mol, KAS: 104 kJ/mol, and Starink: 104 kJ/mol). In addition, the Coats–Redfern and Criado non-isoconversional models were used to identify the most appropriate reaction mechanism of PLA pyrolysis. Accordingly, the best controlling reaction mechanism of the PLA pyrolysis was the geometrical contraction model (R2).

The obtained kinetic parameters (*E*, *A*, and the reaction mechanism) are of great importance for the reactor design. However, it is highly recommended that the type of pyrolysis products together with their caloric values be determined as soon as possible.

## Figures and Tables

**Figure 1 polymers-15-00012-f001:**
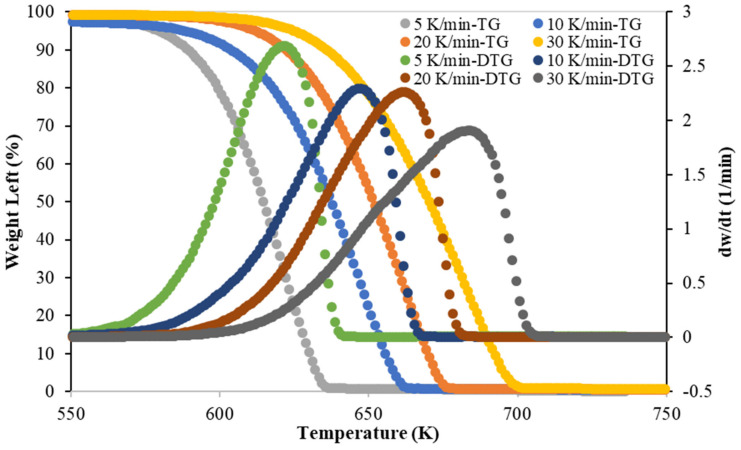
Thermogravimetric analysis (TGA) and Derivative thermogravimetric (DTG) curves for pyrolytic cracking of PLA at different heating rates.

**Figure 2 polymers-15-00012-f002:**
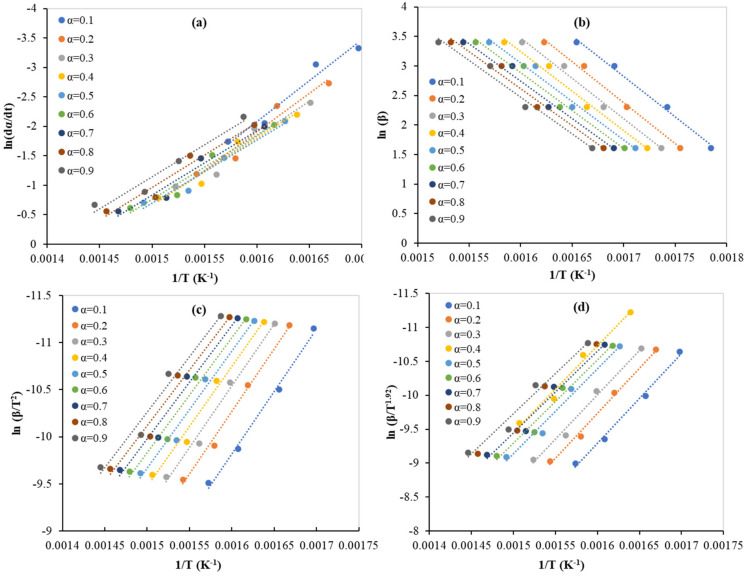
Regression lines of the experimental data of the PLA pyrolysis by: (**a**) Friedman, (**b**) FWO, (**c**) KAS, and (**d**) Starink models.

**Figure 3 polymers-15-00012-f003:**
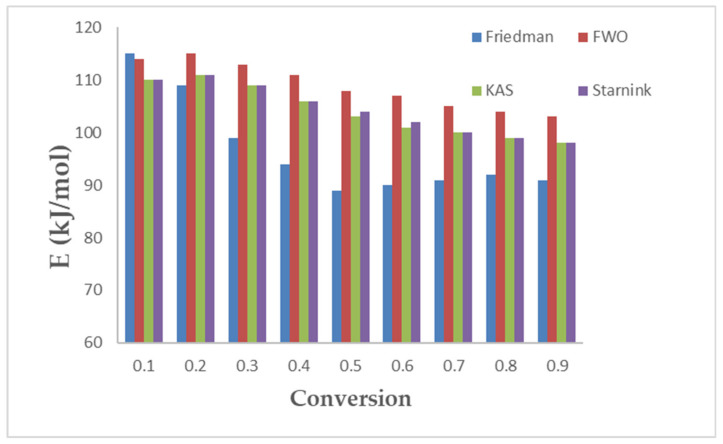
Activation energies for (Friedman, FWO, KAS, and Starink) of PLA pyrolysis.

**Figure 4 polymers-15-00012-f004:**
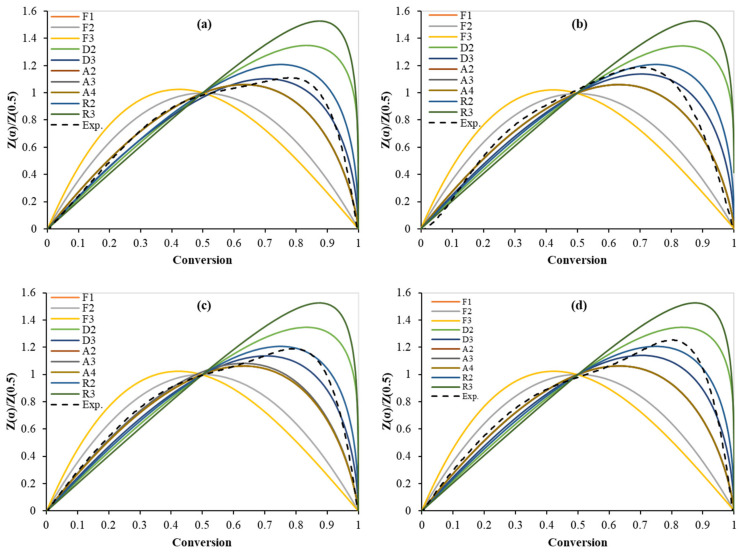
Criado model master plots of the PLA pyrolysis at different heating rates: (**a**) 5 K/min (**b**) 10 K/min, (**c**) 20 K/min, and (**d**) 30 K/min.

**Table 1 polymers-15-00012-t001:** Some of the PLA and different polymers pyrolysis research work.

Polymer/Composite	Findings	Reference
**PLA,** **PLA/nano silver, PLA/nanoclay**	The E values increased when the nano-additives were added. The E value of pure PLA was reported as 164 ± 24 kJ/mol. Incorporating 1 wt% of nano-silver and nano-clay into PLA film shifted the decomposition temperatures. The E values of the composites of PLA/nano-silver and PLA/ nano-clay were reported as 207 ± 24 kJ/mol and 193 ± 24 kJ/mol, correspondingly.	Mróz et al. (2013) [12]
**PLA,** **PLA/PEG,** **PLA/MCC * composites**	One thermal decomposition stage, ranging between 200 and 380 °C, for PLA, PLA/PEG, and PLA/MCC composites was reported. The E value of the pyrolysis of pure PLA was reported as 79.21 KJ/mol and it was reduced to 65 KJ/mol when 10 wt% of PEG was added to PLA. PLA has higher thermal stability when compared to PLA-based composites. The kinetics study revealed that the addition of PEG and MCC to PLA decreases the E value, suggesting resistance reduction for the degradation process.	Bhiogade et al. (2020) [13]
**PP **/PLA blends**	Each polymer in the blends underwent two stages of degradation. The E value was reported as 140 ± 1 kJ/mol for the PP/PLA blend. The other kinetic triplets for the blend were obtained as well.	Hayoune et al. (2020) [14]
**PLA/PBAT *** blends**	TGA curves for mixtures of PLA and PBAT exhibited two-step weight loss profiles.	Xiang et al. (2020) [15]
**Polystyrene (PS)**	The E value of pure PS ranged between 236–286 kJ/mol. Degradation of pure PS took place in the range of 330–380 °C. Pyrolysis data obtained via FWO method.	Mortezaeikia et al. (2021) [4]
**PLA**	The E value of pure PLA reported as 161.75 kJ/mol. Pyrolysis study performed at 5,10 and 15 K/min heating rate under N_2_ atmosphere with flow rate of 50 mL/min. Kinetic study of PLA pyrolysis reported using Friedman method.	Mortezaeikia et al. (2021) [4]
**LDPE**	Degradation occurs at temperature ranges 460–492 °C under N_2_ atmosphere. Degradation occurs at temperature ranges 465–494 °C under CO_2_ atmosphere. LDPE required more energy (446.7 kJ/mol) to initiate the reaction than under N_2_ compared to CO_2_ (467 kJ/mol) atmospheres.	Saad et al. (2021) [5]
**Polyethylene (PE)**	Pyrolysis was carried out using thermogravimetric experimental data at heating rates from 5 to 20 K/min under an argon atmosphere. The activation energy of pure PE was determined by KAS, Friedman and AIC methods as 561.55, 252.58 and 261.86 kJ/mol, respectively.	Pan et al. (2021) [6]
**Lychee and plastic wastes (LPW)**	A co-pyrolysis of LPW can increase the output of bio-oil, reduce carbon coking, improve profitability, and enable industrial production. Study conducted at four different heating rates: 10 °C/min, 20 °C/min, 30 °C/min, and 40 °C/min. Thermal degradation of lychee and plastic waste was 83% using co-pyrolysis. Thermo-kinetic parameters and thermodynamic measurements showed the conversions enthalpies (58–65 kJ/mol) and activation energies (67–71 kJ/mol).	Zhang et al. (2022) [7]

* microcrystalline cellulose. ** polypropylene. *** poly (butylene adipate-co-terephthalate).

**Table 2 polymers-15-00012-t002:** Some physical properties of the PLA samples.

Property	Value
Specific Gravity	1.24
Clarity	Transparent
Tensile Strength @ Break (MPa)	53
Tensile Yield Strength (MPa)	60
Tensile Modulus (GPa)	3.5
Tensile Elongation (%)	6.0
Melting Temperature (°C)	145–160
Glass Transition Temperature (°C)	55–60

**Table 3 polymers-15-00012-t003:** Proximate analysis of PLA.

Moisture	Volatile	Ash
0.6	99.4	0

**Table 4 polymers-15-00012-t004:** Isoconversional models (Mortezaeikia et al. (2021) [4], Mumbach et al. (2019) [20]).

Model	Equation	Comment
Friedman	$\ln\left(\beta \frac{d\alpha }{dT}\right)=\ln\left(A\;f(\alpha \right))-\frac{E}{R\;T}$	The logarithm of Equation (2)
Flynn-Wall-Qzawa (FWO)	$\ln\beta =ln\frac{A\;E}{\;R\;g\left(\alpha \right)}-5.331-1.052\;\;\frac{E}{RT}$	Using Doyle’s approximation
Kissinger-Akahira-Sunose (KAS)	$\ln\frac{\beta }{T^{2}}=ln\frac{A\;R}{\;E\;g\left(\alpha \right)}-\frac{E}{RT}$	Using Murry-White approximation
Starink	$\ln\frac{\beta }{T^{1.92}}=ln\frac{A\;R^{0.92}}{\;E_{a}{}^{0.92}\;g\left(\alpha \right)}-0.312-1.0008\;\frac{E}{RT}$	Using Starink approximation

**Table 5 polymers-15-00012-t005:** Non-isoconversional models. (Mumbach et al. (2019) [20], Aboulkas et al. (2010) [17]).

Model	Equation	Comment
Coats-Redfern	$\ln\frac{g\left(\alpha \right)}{T^{2}}=ln\frac{A\;R}{\;E\;\beta }-\frac{E}{RT}$	Applies an asymptotic series expansion.
Criado *	$\frac{Z\left(\alpha \right)}{Z\left(0.5\right)}={\left(\frac{T_{\alpha }}{T_{0.5}}\right)}^{2}\;\frac{{\left(\frac{d\alpha }{dt}\right)}_{\alpha }}{{\left(\frac{d\alpha }{d\alpha }\right)}_{0.5}}$	Combination of Equation (1) and Coats-Redfern equation.

* The subscript of 0.5 refers to the condition at which α = 0.5.

**Table 6 polymers-15-00012-t006:** Onset, maximum, and end temperatures of pyrolytic cracking characteristics of PLA at different heating rates.

Run No.	Heating Rate (K/min)	T_onset_ (K)	T_peak_ (K)	T_endset_ (K)
1	5	534	623	642
2	10	543	647	668
3	20	566	663	682
4	30	580	684	707

**Table 7 polymers-15-00012-t007:** Peak temperatures for the PLA pyrolysis at 10 K/min.

Ref.	This Work	Bhiogade et al. (2020) [13]	Mróz et al. (2013) [12]	Hayoune et al. (2022) [14]	Xiang et al. (2020) [15]
**T_max_ (K)**	647.0	638	625.3	622	657

**Table 8 polymers-15-00012-t008:** Activation energy of PLA pyrolysis obtained from isoconversional models.

Conversion	Friedman		FWO		KAS		Starink	
	*E* (kJ/mol)	*R^2^*	*E* (kJ/mol)	*R^2^*	*E* (kJ/mol)	*R^2^*	*E* (kJ/mol)	*R^2^*
0.1	115	0.9635	114	0.9944	110	0.9931	110	0.9932
0.2	109	0.9533	115	0.9944	111	0.9932	111	0.9933
0.3	99	0.9505	113	0.9904	109	0.9885	109	0.9886
0.4	94	0.9543	111	0.9870	106	0.9842	106	0.9842
0.5	89	0.9675	108	0.9843	103	0.9809	104	0.9811
0.6	90	0.9536	107	0.9824	101	0.9784	102	0.9786
0.7	91	0.9574	105	0.9809	100	0.9766	100	0.9768
0.8	92	0.9513	104	0.9794	99	0.9746	99	0.9748
0.9	91	0.9608	103	0.9777	98	0.9724	98	0.9726
**Average**	**97**	**0.9569**	**109**	**0.9857**	**104**	**0.9824**	**104**	**0.9826**

**Table 9 polymers-15-00012-t009:** Reported values of activation energies of the PLA pyrolysis.

References	*E* (kJ/mol)	Method
McNeill and Leiper (1985) [27]	119	
Aoyagi et al. (2002) [25]	80–160	
Li et al. (2009) [26]	166.0 ± 6.3 174.7 ± 16.5	FWO IKP + Coats–Redfern
Bhiogade et al. (2020) [13]	79.21	Coats–Redfern + Mampel
**This work**	97 ± 8.76	Friedman
	109 ± 4.25	FWO
	104 ± 4.72	KAS
	104 ± 4.64	Starink

**Table 10 polymers-15-00012-t010:** Activation energy values obtained by the Coats–Redfern model.

Reaction Mechanism	Heating Rates (K/min)								Average	
	5		10		20		30			
	*E*	R^2^	*E*	R^2^	*E*	R^2^	*E*	R^2^	*E*	R^2^
	(kJ/mol)		(kJ/mol)		(kJ/mol)		(kJ/mol)		(kJ/mol)	
F1	256	0.9972	185	0.9995	197	1	177	0.9998	204	0.9991
F2	533	0.983	253	0.9954	257	0.9996	274	0.9982	329	0.9941
F3	899	0.9737	334	0.9895	327	0.9987	393	0.9964	488	0.9896
D1	206	0.9938	271	0.9988	303	0.9993	223	0.9995	251	0.9979
D2	278	0.9985	303	0.9997	333	0.9996	263	0.9999	294	0.9994
D3	392	0.9998	341	1	368	0.9999	313	1	354	0.9999
A2	123	0.997	87	0.9995	93	1	83	0.9998	97	0.9991
A3	79	0.9967	55	0.9994	58	1	52	0.9997	61	0.999
A4	56	0.9964	38	0.9994	41	1	36	0.9997	43	0.9989
R1	98	0.9931	4	0.9996	146	0.9992	106	0.9994	89	0.9978
R2	163	0.9999	4	0.9999	170	0.9998	139	1	119	0.9999
R3	191	0.9998	4	0.9999	179	0.9999	151	1	131	0.9999
P2	44	0.9914	60	0.9984	68	0.9991	47	0.9992	55	0.997
P3	26	0.9888	36	0.998	41	0.9989	28	0.999	33	0.9962
P4	17	0.985	25	0.9975	28	0.9987	18	0.9987	22	0.995

**Table 11 polymers-15-00012-t011:** Kinetic parameters of the PLA pyrolysis obtained by the Coats–Redfern model.

Run No.	Heating Rate (K/min)	*E* (kJ/mol)	*ln(A)*	R^2^	Reaction Mechanism
1	5	163	29.33	0.9999	Geometrical contraction model (R2)
2	10	119	24.56	0.9999	Geometrical contraction model (R2)
3	20	170	30.12	0.9998	Geometrical contraction model (R2)
4	30	139	23.75	1.0000	Geometrical contraction model (R2)

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
