# Peer review of "Pyrolysis Kinetic Study of Polylactic Acid"

_polymers, 2022, doi:10.3390/polym15010012_

Round 1

Reviewer 1 Report

Presented manuscript describes the kinetic aspects of polylactide pyrolysis. The authors used a range of kinetic approaches to analyze the process. My comments from the beginning:

Introduction and overall comment to kinetic analysis - please consult and discuss the latest overview and recommendations for pyrolysis kinetics https://doi.org/10.3390/thermo2040029

"ICATAC's kinetics committees strongly recommend that multiple heating rates be used to achieve more reliable kinetic parameters. [5]" - should be ICTAC. Ref.5 is referred not to the ICTAC recommendations, please refer most recent recommendations on this topic https://doi.org/10.1016/j.tca.2020.178597 https://doi.org/10.1016/j.tca.2022.179384

Table 4 lists well-known theoretical reaction models, it should be moved to the Supplementary material

"Researchers utilized different numerical methods and series expansions to approxi-mate the polynomial term [P(−??? )] and thus different models were proposed." - please provide the related references

Figure 2 reports the same data as Table 9, and it should be moved to the Supplementary material

When comparing the output of the different isoconversional methods, please discuss its different accuracy and averaging over the 0..alpha regions (relevant to KAS, FWO, and not applicable to Friedman), see https://doi.org/10.3390/molecules24122298

"method “Coats-Redfern”" should be just Coats-Redfern method with a proper reference https://doi.rg/10.1038/201068a0

https://doi.org/10.1038/201068a0

Reviewer 2 Report

This manuscript studies the pyrolysis of the polylactic acid using kinetic modeling based on the thermogravimetric analysis results. The subject is interesting and the manuscript has potential. However, the manuscript needs revisions before final decision.

1. Please avoid using abbreviations in the title.

2. The literature survey presented in Table 1 is not enough and up-to-date. Only three references belong to 2020 and the others are very old. It is strongly recommended to expand and update this list. The authors can present similar publications on other polymers, for instance, you can use this review https://doi.org/10.1016/j.jaap.2021.105340 or these research papers https://doi.org/10.1016/j.jaap.2021.105135, https://doi.org/10.1016/j.energy.2022.124678, https://doi.org/10.1016/j.ces.2022.118146, https://doi.org/10.1007/s12649-020-01181-4.

3. What are the novelties of this study compared to the literature? The novelties and the contributions should be clearly highlighted in the last paragraph of the Introduction. In its current format, only a summarized methodology has been described.

4. Please provide the supporting reference for Equation (1), my recommendation is https://doi.org/10.1007/s13762-021-03535-4.

5. Please provide the supporting references for Table 4.

6. Please provide the supporting references for Tables 5 and 6.

7. “TG and DTG at four different heating rates (5, 10, 20, and 30 K/min) of pyrolytic cracking of PLA in Figure 1”. This sentence does not have verb.

Round 2

Reviewer 2 Report

This manuscript studies the pyrolysis of the polylactic acid using kinetic modeling based on the thermogravimetric analysis results. The authors have addressed most of my previous comments. However, some of my comments have not been addressed throughout the manuscript. The literature survey has been improved by recent publications; however, there is not a systematic layout for presenting the literature survey and they are only presented. The authors can present them by subject categories or from the oldest one to the newest one. Also, I recommended some publications in 2022 in the previous stage. Please use the recent publications of 2022 and 2023 and make a systematic layout for presenting the literature.

Also, I recommended to provide the supporting reference for Equation (1). It was only as an example. Other equations such as Equations (2)-(4) need also supporting references. Also, I recommended a recent publication. The authors added a reference from 2010 and stated that “The above reference is quality control techniques which is slightly far from our work”. That reference is on plastic waste pyrolysis by utilizing quality control techniques and is a potential reference for supporting the equations. Instead of using the old references, please provide the recent potential publications regardless the type of the research. All of the previous publications have significant differences with your study otherwise your study has no novelties. Therefore, it is strongly recommended to update you references where it is possible.

Author Response

Please check the attached file as PDF

Round 3

Reviewer 2 Report

This manuscript studies the pyrolysis of the polylactic acid using kinetic modeling based on the thermogravimetric analysis results. The authors have addressed my previous comments and modified the manuscript accordingly. Therefore, I believe that the manuscript deserves the publication in its current format.